

# Remember the past: A comparison of time-adaptive training schemes for non-homogeneous regression

Moritz N. Lang[1,2], Sebastian Lerch[3], Georg J. Mayr[2], Thorsten Simon[1,2], Reto Stauffer[1,4], and Achim Zeileis[1]

[1]Department of Statistics, Universität Innsbruck, Innsbruck, Austria
[2]Department of Atmospheric and Cryospheric Sciences, Universität Innsbruck, Innsbruck, Austria
[3]Institute for Stochastics, Karlsruher Institut für Technologie, Karlsruhe, Germany
[4]Digital Science Center, Universität Innsbruck, Innsbruck, Austria

**Correspondence:** Moritz N. Lang (moritz.lang@uibk.ac.at)

**Abstract.** Non-homogeneous regression is a frequently-used post-processing method for increasing the predictive skill of probabilistic ensemble weather forecasts. To adjust for seasonally varying error characteristics between ensemble forecasts and corresponding observations, different time-adaptive training schemes, including the classical sliding training window, have been developed for non-homogeneous regression. This study compares three such training approaches with the sliding-window approach for the application of post-processing near-surface air temperature forecasts across Central Europe. The predictive performance is evaluated conditional on three different groups of stations located in plains, in mountain foreland, and within mountainous terrain, as well as on changes in the ensemble forecast system of the European Centre for Medium-Range Weather Forecasts (ECMWF) used as input for the post-processing.

The results show that time-adaptive training schemes using data over multiple years stabilize the temporal evolution of the coefficient estimates, yielding an increased predictive performance for all station types tested compared to the classical sliding-window approach based on the most recent days only. While this may not be surprising under fully stable model conditions, it is shown that "remembering the past" from multiple years of training data is typically also superior to the classical sliding-window when the ensemble prediction system is affected by certain model changes. Thus, reducing the variance of the non-homogeneous regression estimates due to increased training data appears to be more important than reducing its bias by adapting rapidly to the most current training data only.

## 1 Introduction

The need of accurate probabilistic weather forecasts steadily increases, because reliable information about the expected uncertainty is crucial for optimal risk assessment in agriculture and industry, or for personal planning of outdoor activities. Therefore, most forecast centers nowadays issue probabilistic forecasts based on ensemble prediction systems (EPSs). To quantify the uncertainty of a specific forecast, an EPS provides a set of numerical weather predictions using slightly perturbed initial conditions and different model parameterizations (Palmer, 2002). However, due to various constraints and required simplifications in the EPS, these forecasts often show systematic biases and capture only parts of the expected uncertainty; especially when EPS





forecasts are directly compared to point measurements (Gneiting and Katzfuss, 2014). In order to increase the predictive skill
of the forecasts for specific locations, statistical post-processing is often applied to correct for these systematic errors in the
forecasts' expectation and uncertainty.

One of the most frequently used parametric post-processing methods is 'ensemble model output statistics' (EMOS) in-
troduced by Gneiting et al. (2005). To emphasize that not only the errors in the mean but also the errors in the uncertainty
are corrected, the method is often referred to as 'non-homogeneous regression' (NR). In the statistical literature, this type of
model is also known as distributional regression (Klein et al., 2014) since all parameters of a specific response distribution are
optimized simultaneously conditional on respective sets of covariates.

As the error characteristics between the covariates, typically provided by the EPS, and the observations often show seasonal
dependencies and might change inter-annually over time, different time-adaptive training schemes have been developed for NR
models. Gneiting et al. (2005) proposed the so-called 'sliding training window' approach where the training data set consists of
EPS forecasts and observations of the most recent 30–60 days only. As soon as new data become available, the training data set
and the statistical model are updated so that the estimated coefficients automatically evolve over time and adjust to changing
error characteristics. This makes it very handy for operational use, however, little training data can sometimes yield unrealistic
jumps in the estimated coefficients over time, especially if events which show a significantly different error characteristic
enter the training data set. Therefore, to stabilize the temporal variability of the coefficient estimates, several approaches have
been proposed in literature. Scheuerer (2014) regularizes the estimation by only allowing the optimizer to slightly adjust the
coefficient from day to day. In an alternative approach, Möller et al. (2018) extend the training data by using not only the days
prior to estimation, but also the days centered around the same calendar day over all previous years available. This idea of
using a rolling centered training data set over multiple years is similar to the concept of using annual cyclic smooth functions
to capture seasonality as employed by Lang et al. (2019). These smooth functions are also known as regression splines (Wood,
2017), where the estimate of each point in the function only depends on data in its closer neighborhood; this allows for a
smooth and stable evolution of the coefficients over the year.

Alternative time-adaptive models are based on historical analogs or non-parametric approaches. For approaches employing
analogs (Junk et al., 2015; Barnes et al., 2019), training sets are selected to consist of past forecast cases with atmospheric
conditions similar to those on the day of interest. Such methods may lead to models that are able to account for the flow-
dependency of EPS errors (Pantillon et al., 2018; Rodwell et al., 2018). However, the definition and computation of similarity
measures is far from straightforward, and substantial methodological developments may be required to obtain suitably extensive
training data sets for stable model estimation (Hamill et al., 2008; Lerch and Baran, 2017). For non-parametric approaches
(Taillardat et al., 2016; Henzi et al., 2019) or semi-parametric approaches (Rasp and Lerch, 2018; Schlosser et al., 2019),
time-adaptive choices of the training data are typically abandoned as well, as interactions between the day of the year and
other covariates can capture the potential time-adaptiveness. Therefore, analog-based and non-parametric approaches will not
be pursued further in the context of this work.

In addition to the training scheme employed, an important data-specific aspect which has to be considered in post-processing
is that the EPS may change over time (Hamill, 2018). Changes in the underlying numerical model such as, e.g., an increased





horizontal resolution, can possibly lead to sudden transitions in the predictive performance of the EPS and hence affect the error characteristics of the data. If the training data set used to estimate the statistical post-processing model contains data of a previous EPS version which significantly differs from the current one, it can result in a loss of the predictive performance.

This paper presents a comparison of four different time-adaptive training schemes proposed in literature. A case study is shown for post-processed 2 m temperature forecasts for three different groups of stations across Central Europe in the midlatitudes, namely stations in the plain, in the foreland, and within mountainous terrain (Fig. 1). The study highlights the advantages and drawbacks of the different approaches and investigates the impact of changes in the EPS on the predictive performance in different topographical environments.

The structure of the paper is as follows: Section 2 explains the different methods and the comparison setup including the underlying data. In Sect. 3, the different time-adaptive training schemes are compared in terms of their coefficient paths and their predictive performance. Finally, a summary and conclusion is given in Sect. 4.

## 2   Methodology and comparison setup

The different training schemes for NR models proposed in the literature try to adapt to various kinds of error sources that can occur in post-processing, both in space and time. In order to provide a unifying view and to fix jargon, we first discuss these different error sources and then introduce the training schemes considered along with the comparison setup employed.

### 2.1   Sources of errors in post-processing

NR models aim to adjust for errors and biases in EPS forecasts but, of course, the NR models can be affected by errors and misspecifications themselves. Therefore, we try to carefully distinguish the two different models involved with their associated errors, i.e., the numerical weather prediction model underlying the EPS vs. the statistical NR model employed for post-processing.

The skill of the EPS can be quantified in EPS forecast biases and variances which (i) typically vary for different locations conditional on the surrounding terrain, (ii) often show cyclic seasonal patterns, and (iii) can experience non-seasonal temporal changes, e.g., due to changes in the EPS itself.

In addition to the error sources in the employed EPS, the performance of the statistical post-processing itself will typically also (iv) differ at different measurement sites, (v) strongly depend on the amount of training data used, and (vi) whether it is affected by effects that are not accounted for in the NR specification.

Clearly, larger training samples (v) will lead to more reliable predictions when the NR specification (vi) – in terms of response distribution, covariates and corresponding effects, link functions, estimation method, etc. – appropriately captures the error characteristics in the relationship between EPS forecasts and actual observations. However, when these error characteristics differ in space (i and iv) and/or in time (ii and iii), it is not obvious what the best strategy for training the NR is. Extending the training data (v) in space or time will reduce the variance of the NR estimation but might also introduce bias if the NR





specification (vi) is not adapted. Thus, this is a classical bias-variance trade-off problem and we investigate which strategies
for dealing with this are most useful in a typical temperature forecasting situation.

To fix jargon, we employ the terms "model" and "bias" without further qualifiers when referring to the NR model in post-processing. Whereas when referring to the numerical weather prediction model we employ "EPS model" and "EPS bias". Moreover, we refer to a statistical model whose estimates have small bias and variance as stable.

### 2.2 Non-homogeneous regression with time-adaptive training schemes

Non-homogeneous regression as originally introduced by Gneiting et al. (2005) is a special case of distributional regression, where a response variable $y$ is assumed to follow a specific probability distribution $\mathcal{D}$ with distribution parameters $\theta_k, k = 1, \ldots, K$:

$$y \sim \mathcal{D}(\theta_1, \ldots, \theta_K) = \mathcal{D}(h_1(\eta_1), \ldots, h_K(\eta_K)), \tag{1}$$

where each parameter of the distribution is linked to an additive predictor $\eta_k$ via a link function $h_k$ to ensure its appropriate
co-domain. In case of post-processing air temperatures, the normal distribution is typically employed (Gneiting and Katzfuss, 2014), and Eq. (1) can be rewritten as

$$y \sim \mathcal{N}(\mu, \sigma). \tag{2}$$

In the classical NR (Gneiting et al., 2005), the two distribution parameters location $\mu$ and scale $\sigma$ are expressed by the ensemble mean $m$ and ensemble variance or standard deviation $s$, respectively:

$$\mu = \eta_\mu = \beta_0 + \beta_1 \cdot m, \tag{3}$$

$$\log(\sigma) = \eta_\sigma = \gamma_0 + \gamma_1 \cdot s, \tag{4}$$

with $\beta_\bullet$ and $\gamma_\bullet$ being the corresponding intercept and slope coefficients. Here, we use the logarithm link to ensure positivity of the scale parameter $\sigma$, however, a quadratic link with additional parameter constraints for the coefficients as used by Gneiting et al. (2005) would also be feasible. In this study, we regard the statistical model specifications according to Eq. (2)–(4), but
all concepts of time-adaptive training schemes could easily be transferred to other response distributions $\mathcal{D}$, to alternative link functions $h(\cdot)$, or to more complex additive predictors $\eta$ with additional covariates.

The regression coefficients $\beta_\bullet$ and $\gamma_\bullet$ are estimated by minimizing a loss function over a training data set containing historical pairs of observations and EPS forecasts. In this study, we employ maximum likelihood estimation, which performs very similar to minimizing the continuous ranked probability score (CRPS, Gneiting and Raftery 2007) as used by Gneiting et al. (2005)
when the response distribution is well specified (Gebetsberger et al., 2018). For a single observation $y$, the log-likelihood $L$ of the normal distribution is given by

$$L(\mu, \sigma | y) = \log \left\{ \frac{1}{\sigma} \phi \left( \frac{y - \mu}{\sigma} \right) \right\}, \tag{5}$$





where $\phi(\cdot)$ is the probability density function of the normal distribution. The coefficients $\beta_\bullet$ and $\gamma_\bullet$, specified in Eq. (3) and (4), are derived by minimizing the sum of negative log-likelihood contributions $L$ over the training data. The larger the training

data the more stable is the estimation in case the statistical model is well specified; however, if the covariate's skill varies either seasonally or non-seasonally over time, this leads to the bias-variance trade-off between preferable large training data sets for stable estimation and the benefit of shorter training periods which allow to adjust more rapidly to changes in the data or, to be precise, in the error characteristics of the data (see Sect. 2.1). In the following, four approaches are discussed how to gain informative time-adaptive training data sets while ensuring a stable estimation.

### 2.2.1   Sliding-window

The *sliding-window* approach originally introduced by Gneiting et al. (2005) uses the most recent days prior to the day of interest as training data for estimation. For post-processing 2 m temperature forecasts, Gneiting et al. (2005) found the best predictive performance for training periods between 30 and 45 days with substantial gains in increasing the training period beyond 30 days and slow but steady performance losses for training lengths beyond 45 days. According to Gneiting et al.

(2005), the latter is presumably a result of seasonally varying EPS forecast biases.

In this study, we use a period of 40 days for the *sliding-window* approach, which is a frequently used compromise (e.g., Baran and Möller 2017; Gneiting et al. 2005; Wilson et al. 2007). However, as discussed in Gneiting et al. (2005), different training periods might perform better for distinct weather variables, locations, forecast steps, or model specifications. Common choices in the literature include training lengths between 15 and 100 days, for example depending on whether the estimation

of regression coefficients is performed station-specific or jointly for multiple locations at once.

### 2.2.2   Regularized sliding-window

A regularized adaption of the classical *sliding-window* approach was introduced by Scheuerer (2014) in order to stabilize the estimation based on early stopping in statistical learning. The motivation is that gradient-based optimizers adjust the starting values by iteratively taking steps in the direction of the steepest descent of a distinct loss function until some convergence

condition is fulfilled. These steps are largest in the first iteration and getting smaller towards the optimum. Thus, the most important adjustments are made during the first steps, while further adjustments often improve the fit to unimportant or even random features in the data which can lead to wiggly coefficient paths over time and ultimately to an overfitting (Scheuerer, 2014).

Therefore, Scheuerer (2014) proposes to use the coefficients of the previous day as starting values and to stop the optimizer

after a single iteration to stabilize the evolution of the coefficient estimates. A drawback of his approach is that it implies that the estimation never converges and in case of poor starting values or strong truly observed temporal changes in the data the obtained coefficients might be incorrect (Scheuerer, 2014). For post-processing precipitation, Scheuerer (2014) obtained better results with regularized coefficients than without regularization.

For the *regularized sliding-window* approach used in this study, we employ the quasi-Newton Broyden-Fletcher-Goldfarb-

Shanno (BFGS) algorithm as in Scheuerer (2014) and stop the optimizer after one single iteration. For the first time, we let





the BFGS algorithm perform 10 iterations and use $(\beta_0, \beta_1)^\top = (0, 1)^\top$ as starting values in the location parameter $\mu$ and $(\gamma_0, \gamma_1)^\top = (0.1, 1)^\top$ as starting values in the scale parameter $\sigma$. In comparison to Scheuerer (2014), we perform maximum likelihood estimation instead of CRPS minimization.

### 2.2.3   Sliding-window plus

As already pointed out by Gneiting et al. (2005), training data from previous years could additionally be included in the *sliding-window* approach to address seasonal effects. This should reduce the variance in the estimation of the regression coefficients, which stabilizes the evolution of the coefficients similar to the *regularized sliding-window* approach.

This idea has recently been pursued by Vogel et al. (2018) for the construction of climatological reference forecasts, and by Möller et al. (2018) for a post-processing approach based on D-vine copulas in which much more coefficients than in classical

NR need to be estimated, making a more extensive training data set necessary. Their so-called 'refined training data set' consists of the most 45 recent days prior to the day of interest, plus 91 days centered around the same calendar day over all previous years available. Including multiple years yields more stable estimates while, on the other hand, there is the trade-off of losing the ability to quickly adjust to non-seasonal temporal changes in the EPS forecast biases. The approach of Möller et al. (2018) can be seen as time-adaptive version of the seasonal training proposed by Hemri et al. (2016) who consider training data sets

comprised of days from all previous years within the same season (winter/summer) as the day of interest.

In this study, to be comparable to the *sliding-window* approach we use the most recent 40 days prior to estimation and a respective 81 days interval centered around the day of interest over the previous years available in the training data.

### 2.2.4   Smooth model

If we reformulate the *sliding-window plus* approach, it is very similar to fitting an annual cyclic smooth function where the

points of the function only depend on data points in the closer neighborhood, specified by the sliding window length.

Cyclic smooth functions belong to the broader model class of generalized additive models (GAMs, Hastie and Tibshirani, 1986), which allow one to include potentially nonlinear effects in the linear predictors $\eta$. Smooth functions are also referred to as regression splines and are directly linked to the model parameters as additive terms in $\eta$. Introductory material for cyclic smooth functions conditional on the day of the year can be found in Lang et al. (2019) and a comprehensive summary on

GAMs is given in Wood (2017).

To account for seasonal variations we only need to fit one single model, here called *smooth model*, over a training data set with several years of data. The effects included allow the coefficients to smoothly evolve over the year, which leads to the following adaptions in Eq. (3) and (4) for the location $\mu$ and scale $\sigma$, respectively:

$$\mu = \eta_\mu = \beta_0 + f_0(\text{doy}) + (\beta_1 + f_1(\text{doy})) \cdot m, \tag{6}$$

$$\log(\sigma) = \eta_\sigma = \underbrace{\gamma_0 + g_0(\text{doy})}_{\substack{\text{seasonally varying} \\ \text{intercept}}} + \underbrace{(\gamma_1 + g_1(\text{doy}))}_{\substack{\text{seasonally varying} \\ \text{slope}}} \cdot s, \tag{7}$$





**Table 1.** Overview of time-adaptive training schemes, distinguished by model specification/estimation and training data selection corresponding to errors sources (vi) and (v), respectively. The basic model specification refers to Eq. (3)–(4) in contrast to the extended Eq. (6)–(7).

| Name | Model | | Data | |
| | Specification | Estimation | Years | Seasons |
| --- | --- | --- | --- | --- |
| Sliding-window | Basic | Maximum likelihood | Current | Current |
| Regularized sliding-window | Basic | Early stopping | Current | Current |
| Sliding-window plus | Basic | Maximum likelihood | Multiple | Current |
| Smooth model | Extended | Penalized | Multiple | All |

with $m$ and $s$ being the ensemble mean and ensemble standard deviation, respectively; $\beta_\bullet$ and $\gamma_\bullet$ are regression coefficients, and $f_\bullet(\mathrm{doy})$ and $g_\bullet(\mathrm{doy})$ employ cyclic regression splines conditional on the day of the year (Wood, 2017). The regression coefficients $\beta_0$ and $\gamma_0$, and $\beta_1$ and $\gamma_1$ are unconditional on the day of the year and can be interpreted as global intercept or slope coefficients, respectively.

## 2.3 Comparison setup

### 2.3.1 NR training schemes

The NR training schemes presented in Sect. 2.2 deal with the potential temporal error sources from Sect. 2.1 in different ways (see Table 1 for an overview). The classic *sliding-window* employs the basic NR model equations from Eq. (3)–(4) and avoids potential biases in the NR model estimation by using only very recent data from the same year and season. Compared to this, the *regularized sliding-window* and *sliding-window plus* both try to stabilize the coefficient estimates by reducing the variance – either through regularized estimation (vi) or by considering multiple years (v). The *smooth model* differs from all of these by modifying both the model (vi) and data (v) specification, using the extended model specification from Eq. (6)–(7) fitted by penalized estimation to a large data set comprising several years and all seasons.

Potential spatial differences (i) and (iv) are handled for all training schemes in the same way: The NR models are estimated separately for each station and subsequently evaluated in groups of terrain types (plain, foreland, alpine). The underlying EPS data – described subsequently – is the same for all NR training schemes and thus affected by the same seasonal (ii) and non-seasonal changes (iii).

### 2.3.2 Data sets

For validation of the training schemes we consider 2 m temperature ensemble forecasts and corresponding observations at 15 measurement sites located across Austria, Germany, and Switzerland. The sites are chosen to investigate the impact of potential error sources in space (i) and (iv), e.g., through varying discrepancies between the real and the EPS topography. The data comprises three groups of five stations located either in plains, mountain foreland, and within mountainous terrain (see



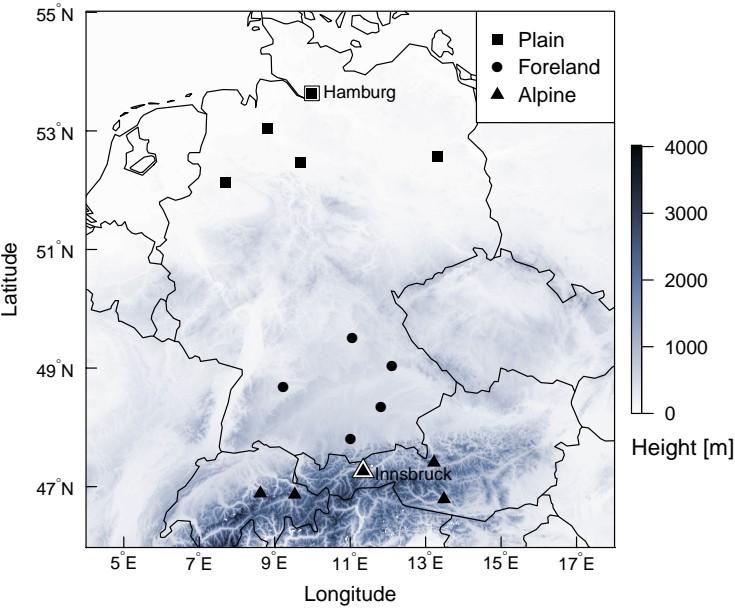

**Figure 1.** Overview of the study area with selected stations classified as plain, foreland, and alpine station sites. The two highlighted and labeled stations, Hamburg and Innsbruck, are discussed in detail in Sect. 3.1. Elevation data are obtained from the SRTM-30 m digital elevation model (NASA JPL, 2013).

Fig. 1). The estimated statistical models for the stations Hamburg and Innsbruck, highlighted by symbols with white borders, are discussed in more detail in Sect. 3.1.

As covariates for Eq. (3)–(7), we employ the ensemble mean $m$ and the ensemble standard deviation $s$ of bilinearly interpolated 2 m temperature forecasts issued by the global 50-member EPS of the European Centre for Medium-Range Weather Forecasts (ECMWF). We assess forecast steps from $+12$ h to $+72$ h ahead on a 12 hourly temporal resolution for the EPS run initialized at 0000 UTC and use data from March 8, 2010 to March 7, 2019.

This period has been selected in order to investigate the impact of non-seasonal long-term changes in the EPS model (iii) that 210 is not reflected in the NR model specifications. Namely, the horizontal resolution of the ECMWF EPS changed on March 8, 2016 (cycle 41r2). Specifically, it is of interest how the *sliding-window plus* and the *smooth model* are affected if the training period comprises data from both the 'old EPS version' before the change as well as the 'new EPS version'. Thus, we construct three data sets with different validation period that are either (A) not affected by the EPS model change at all, (B) start immediately after the model change, or (C) with some time lag after change.

To understand how this affects the different training schemes, we first illustrate in Figure 2a how training and validation period are selected for each scheme. For the three sliding-window approaches, the NR models are re-estimated every day as the validation date rolls through the validation period (hatched area). In contrast, the *smooth model* is estimated only once

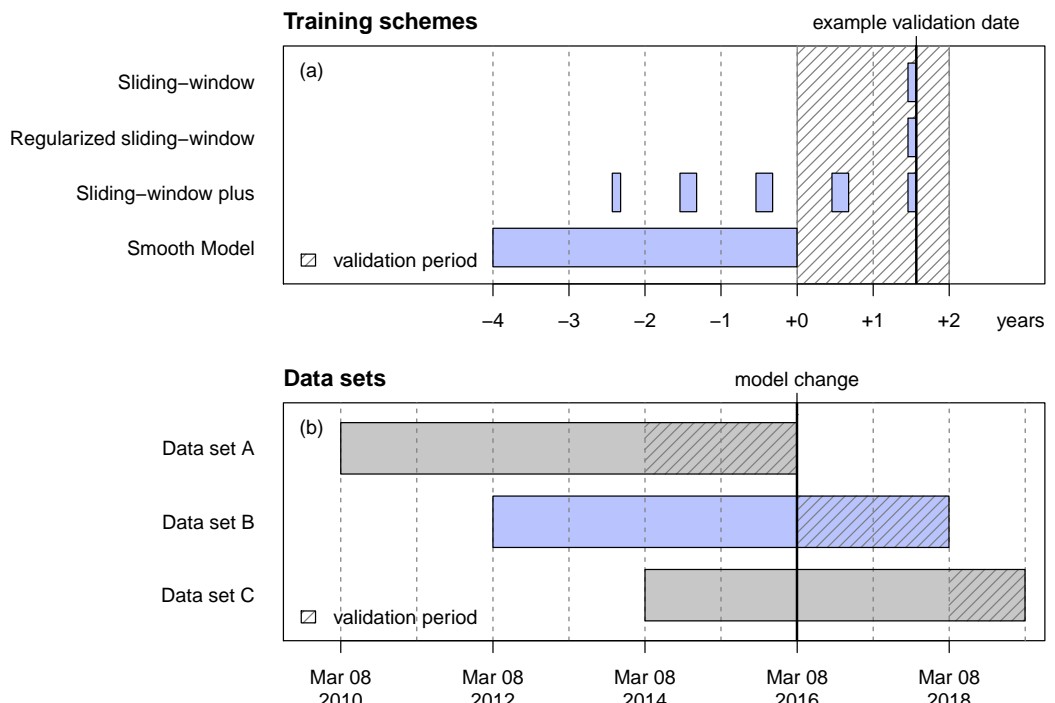

**Figure 2.** (a) Illustrative example of how the training data sets are composed for the four different time-adaptive training schemes. (b) Schematic overview of the training and validation data sets employed in this study with regard to the change in the horizontal resolution of the ECMWF EPS on March 8, 2016 (cycle 41r2). For training, up to four years of data are used in all data sets; for validation, two years of data are used for data sets A and B, and one year for data set C.

for the entire validation period based on a fixed training data period of four years prior to the validation period. For a fair comparison, the training data for the *sliding-window plus* model is also restricted to four years prior to each validation date.

Now Fig. 2b illustrates how the three data sets A, B, and C are selected in relation to the EPS change on March 8, 2016:

  – Data set A: All models are trained and evaluated without being affected by the EPS change.

  – Data set B: All models start with a training period entirely before the EPS change but a validation period entirely after the change. However, for the *sliding-window* and *regularized sliding window* approaches the training period quickly rolls across the change point and after 40 days they are not affected by it anymore. For the *sliding-window plus* the training
data also rolls into the new EPS version but still partially uses data from the old EPS version. Finally, as the *smooth model* is only estimated once it cannot adapt at all to the new EPS version.

  – Data set C: Effects from A and B are mixed so that the *smooth model* and the *sliding-window plus* model use data from both the old and new EPS version, while the classical *sliding-window* and *regularized sliding-window* models already use only data from the new EPS version.





The validation period is 2 years for A and B and 1 year for C. A total number of 731/730/365 NR models has to be estimated
for the three sliding-window approaches, while only 1/1/1 *smooth model* is required for data sets A/B/C per station and forecast
step.

## 3 Results

This section assesses the performance of the different time-adaptive training schemes. First, the temporal evolution of the
estimated coefficients are shown for two stations representative for one measurement site in the plains and one in mountainous
terrain. Afterwards, the predictive performance of the training schemes is evaluated in terms of the CRPS conditional on the
three data sets with and without EPS changes (Fig. 2) and grouped for stations classified as topographically plain, mountain
foreland, and alpine sites (Fig. 1).

### 3.1 Coefficient paths

Figure 3 shows the estimated coefficients for Innsbruck at forecast step $+36\,\mathrm{h}$ conditional on the day of the year. The coeffi-
cient paths are plotted for the different time-adaptive training schemes for two years included in the validation period of data
set A. The pronounced seasonal evolution of the coefficients for all training schemes indicates that the EPS' forecast bias and
skill varies seasonally which makes a time-adaptive training scheme mandatory to capture these characteristics in the post-
processing. Apparently, the EPS temperature forecasts have a higher information content during summer which yields a slope
coefficient $\beta_1$ close to one in the location parameter $\mu$ and a high slope coefficient $\gamma_1$ in the scale parameter $\sigma$ for this period.

In comparison to the other time-adaptive training schemes, the classical *sliding-window* approach (Fig. 3a, d, g, j) shows
very strong outliers and an unstable temporal evolution for all coefficients with distinct differences during the two subsequent
validation years; this is more pronounced for the scale parameter $\sigma$ where the estimates seem to be more volatile than for
the location parameter $\mu$. All strategies extending the classical *sliding-window* approach smooth the temporal evolution of
the coefficients to a certain extent while maintaining the overall seasonal cyclic pattern. For the *regularized sliding-window*
approach (Fig. 3b, e, h, k), the stabilization strongly differs for the individual coefficients and some of the estimated coefficients
seem to need rather long to adapt during the transition periods; the latter could indicate that a single iteration step might not
be sufficient in this study. The coefficient paths for the *sliding-window plus* approach (Fig. 3c, f, i, l) and for the *smooth model*
(Fig. 3a–l; solid line) look very similar with minor distortions during the cold season. Due to the definition of the *smooth
model*, its coefficient paths show the most stable evolution but with the lowest ability to react to abrupt changes in the error
characteristics.

For Hamburg (Fig. 4) by contrast to Innsbruck, the information content of the mean EPS temperature forecast is quite
high throughout the year. This yields a lower bias correction and an almost one-to-one mapping of the ensemble mean to
the location parameter $\mu$ indicated by a coefficient $\beta_1$ close to one. Despite the different post-processing characteristics, the
temporal evolution of the coefficient paths is similar to the one for Innsbruck which confirms our previous findings: For
the extended sliding-window approaches the coefficients have indeed very little seasonal variability, while for the classical

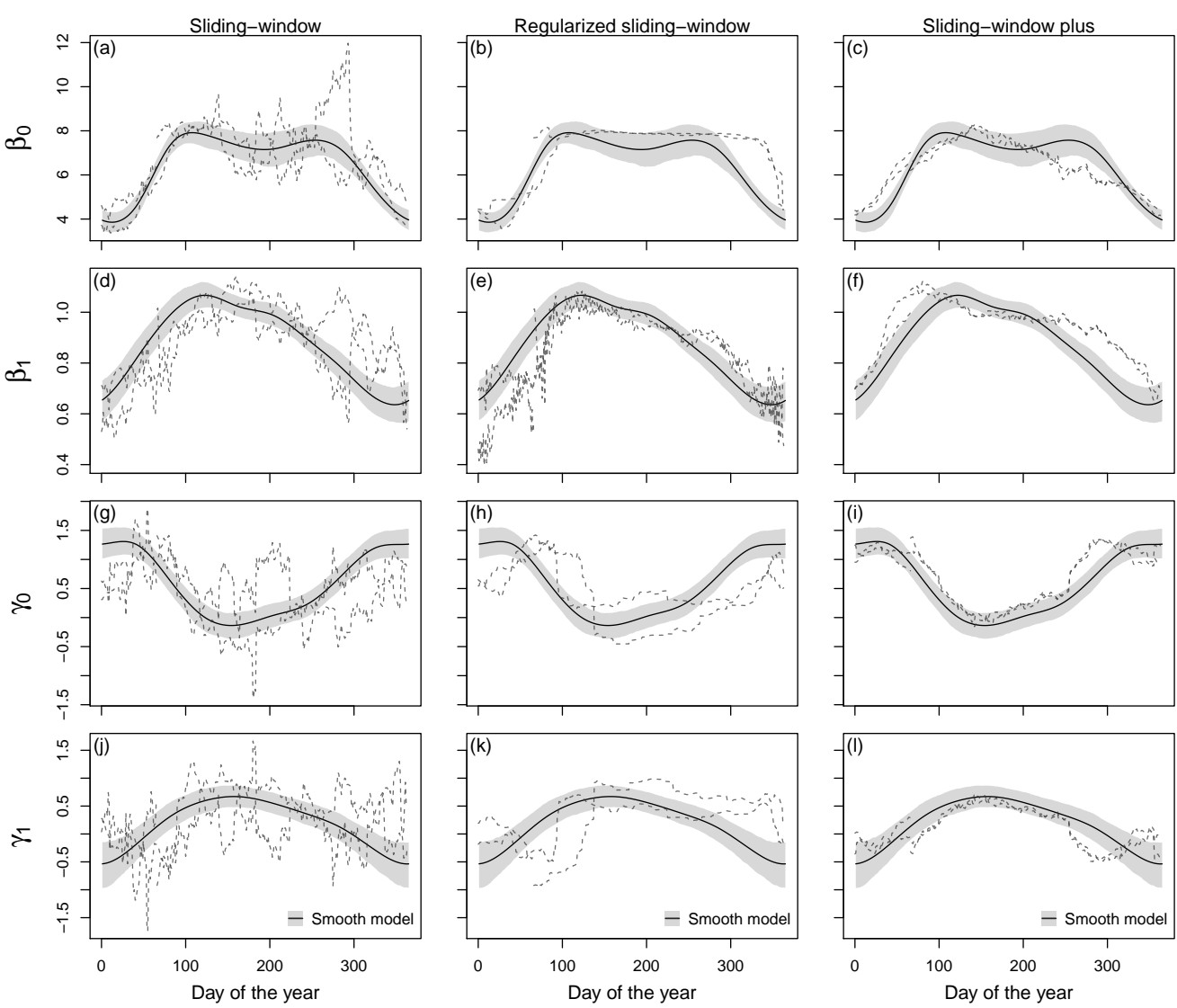

**Figure 3.** Temporal evolution of regression coefficients for the validation period in data set A for Innsbruck at forecast step $+36\,\mathrm{h}$ (valid at 1200 UTC). The coefficient paths are shown for the coefficients $\beta_0$ (a–c) and $\beta_1$ (d–f) in the location parameter $\mu$, and for the coefficients $\gamma_0$ (g–i) and $\gamma_1$ (j–l) in the scale parameter $\sigma$ based on the *sliding-window*, *regularized sliding-window*, and *sliding-window plus* approach (dashed, from left to right) compared to the *smooth model* approach (solid line). The different dashed lines in each subplot correspond to individual years. The grey shading represents the 95% credible intervals of the coefficients in the *smooth model* based on MCMC sampling.

*sliding-window* approach the coefficients show unrealistically strong fluctuations over time without a clear seasonal pattern (Fig. 4a, d, g, j). As for Innsbruck, the *regularized sliding-window* approach has a rather unrealistic stepwise evolution for some coefficients (Fig. 4b, e, h, k). The coefficient paths for the *sliding-window plus* approach (Fig. 4c, f, i, l) and the *smooth model*



**Figure 4.** As Fig. 3, but for Hamburg at forecast step +36 h (valid at 1200 UTC).

(Fig. 4; solid line) look comparable. These results support the bias-variance trade-off that regularizing or smoothing stabilizes the coefficient paths, while loosing the ability to rapidly react to temporal changes in the data.

## 3.2 Predictive performance

After the illustrative evaluation of the coefficients' temporal evolution for the different time adaptive training schemes, Fig. 5 shows aggregated CRPS skill scores for groups of five respective stations classified as topographically plain, mountain foreland,

Nonlinear Processes in Geophysics Open Access
Discussions
EGU



**Figure 5.** CRPS skill scores clustered into groups of stations located in the plain, in the mountain foreland near the Alps, and within mountainous terrain and for the out-of-sample validation periods according to the different data sets: Data set A without an EPS change, data set B with an EPS change in between the training and the validation data sets, and data set C with an EPS change withing training data. Compared are the different time-adaptive training schemes specified in Sect. 2.2 with the classical *sliding-window* approach as a reference; note that 'sliding-window' is abbreviated as SW in the figure. Each box-whisker contains aggregated skill scores over the forecast steps from +12 h to +72 h on a 12 hourly temporal resolution and over five respective weather stations (Fig. 1). Skill scores are in percent, positive values indicate improvements over the reference.

and alpine sites (Fig. 1) regarding the data sets A, B, and C (Fig. 2). In all panels the *regularized sliding-window* approach, the *sliding-window plus* approach, and the *smooth model* is compared to the classical *sliding-window* approach as a reference.





– For data set A, the *regularized sliding-window* approach shows only little improvements for the plain and foreland, and an overall performance loss for alpine stations. By contrast, the *sliding-window plus* and the *smooth model* approaches show distinct improvements over the classical *sliding-window* approach with largest values for alpine sites.

– For data set B at stations in the plains and foreland, the mean predictive skill behaves similarly to data set A, except that the *smooth model* shows a slightly larger variance. For alpine stations, the *regularized sliding-window* approach performs slightly worse than in data set A, while the two approaches using training data over multiple years do no longer outperform the reference.

– For data set C at stations in the plains and foreland, the predictive skill is again similar to data set A with slight performance losses. For alpine stations, the *regularized sliding-window* approach shows even less skill as in data set B, while 280 the two other approaches again outperform the *sliding-window* approach and are on a similar level as in data set A.

The validation of the different time-adaptive training schemes shows that the *sliding-window plus* approach and the *smooth model* perform overall similar and are clearly superior for all station types compared to the classical *sliding-window* approach. However, the *smooth model* shows the highest variance in the predictive performance in case of a change in the EPS, especially 285 in mountainous terrain (data sets B and C); this is likely due to its reduced ability to adapt to temporal changes in the data. Furthermore, the validation shows that the *regularized sliding-window* approach seems to have difficulties in mountainous terrain and yields only minor improvements for plain and foreland sites.

## 4  Summary and conclusion

Non-homogeneous regression (NR) is a widely used method to statistically post-process ensemble weather forecasts. In its 290 original version it was used for temperature forecasts employing a Gaussian response distribution, but over the last decade various statistical model extensions have been proposed for other quantities employing different response distributions or to enhance its predictive performance. When estimating NR models there is always a trade-off between large enough training data sets to get stable estimates and still allowing the statistical model to adjust to temporal changes in the statistical error characteristics of the data. Therefore, different training schemes with specific advantages and drawbacks have been developed 295 as presented in this paper.

The classical *sliding-window* approach has the advantage that no extensive training data set is required which allows the statistical model to adjust itself rapidly to changing forecast biases, for example in case of changes in the EPS. On the other hand, statistical models trained on a small training data set have typically large variance in the estimation of the regression coefficients, which can yield unstable wiggly coefficient paths. Additional regularization allows one to stabilize the evolution 300 of the regression coefficients without losing the simplicity of the classical *sliding-window* approach. However, inappropriate settings of the optimizer as, e.g., unrealistic starting values or insufficient update steps, can quickly lead to incorrect coefficients. The alternative *sliding-window plus* strategy foregoes regularization but stabilizes the coefficients by using an extended training data set which includes data from the same season over several years. Compared to the classical approach the method requires





historical data and partially loses its ability to rapidly adjust to changes in the error characteristics. The last approach presented
in this paper can be seen as a generalization of the *sliding-window plus* approach. Rather than using a training data set centered
around the date of interest, the *smooth model* makes use of all historical data in combination with cyclic regression splines
which allows the coefficients to smoothly evolve over the year.

The differences between the methods presented can be seen in the coefficient paths shown in Fig. 3 and 4. The coefficients
of the classical *sliding-window* approach show strong fluctuations and pronounced peaks throughout the year. Regularization
allows to stabilize the evolution, however, strong step-wise changes in the coefficient paths still occur. The two methods using
data from multiple years perform comparably similar with stable coefficient paths over the year. Figure 5 confirms that more
stable estimates have a positive impact on the predictive performance. The *sliding-window plus* approach and the *smooth model*
show an overall improvement of about 3–5 % (in median) over the classical *sliding-window* approach, while the *regularized
sliding-window* only partially outperforms the *sliding-window* training scheme. In case of a change in the EPS the approaches
using multiple years of training data are still superior to the ones using the most recent days only, even if they technically allow
to adjust to the EPS change more rapidly.

To conclude, all four training schemes shown in this paper have their advantages in particular applications. If only short
periods of training data are available (< 1 year), the classical *sliding-window* approach may already provide sufficiently good
estimates. However, as soon as one has access to longer historical data sets, the approaches using data from multiple years
become superior due to a more stable coefficients' evolution over time which yields an overall improved performance. While
the *sliding-window plus* is a natural extension of the classical *sliding-window* approach and, therefore, can be estimated by the
same software, the *smooth model* approach can be seen as a generalization and only a single model has to be estimated for all
seasons using all available data. The *smooth model* yields, by definition, the smoothest and most stable coefficient paths but
with the lowest ability to adjust itself to a new error characteristic.



*Code availability.* All computations are performed in R 3.6.1 (R Core Team, 2019): The statistical models using a sliding-window approach are based on the R package **crch** (Messner et al., 2016) employing a frequentistic maximum likelihood approach. The statistical models using a time-adaptive training scheme by fitting cyclic smooth functions are fitted with the R package **bamlss** (Umlauf et al., 2018). The package provides a flexible toolbox for distribution regression models in a Bayesian framework; introductory material can be found on http://BayesR.R-Forge.R-project.org/. The computation of the CRPS is based on the R package **scoringRules** (Jordan et al., 2019).

*Author contributions.* This study is based on the PhD work of MNL under supervision of GJM and AZ. The majority of the work for this study was performed by MNL with the support of RS. All the authors worked closely together in discussing the results and commenting on the manuscript.

*Competing interests.* The authors declare that they have no conflict of interest.

*Acknowledgements.* This project was partly funded by the Austrian Research Promotion Agency (FFG, grant no. 858537) and by the Austrian Science Fund (FWF, grant no. P31836). Sebastian Lerch gratefully acknowledges support by the Deutsche Forschungsgemeinschaft (DFG) through SFB/TRR 165 "Waves to Weather". We also thank the Zentralanstalt für Meteorologie und Geodynamik (ZAMG) for providing access to the data.





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
