# Peer review of "Remember the past: A comparison of time-adaptive training schemes for non-homogeneous regression"

_Nonlinear Processes in Geophysics, 2019_

## Referee Comment (RC1) · Anonymous Referee #1 · 8 Oct 2019

This manuscript compares the effect of different schemes to compose training data for statistical post-processing methods (here: non-homogeneous regression) on the performance of the resulting forecasts. It is well written and highly relevant to operational forecasting where availability of reforecast data may be limited and the consequences of changes in the NWP model on forecast calibration must be understood in order to decide whether forecasts from an older NWP model version can be used to fit the parameters defining the post-processing model. This last point is the only one where I feel the manuscript could benefit from a more detailed discussion. Specifically: The CRPS skill scores in Fig. 5 h) suggest that the regularization scheme struggles with the adjustment to the NWP model upgrade and to the annual cycle, but also the SW

plus and the smooth model have an overall neutral effect on skill even though these schemes increase the training sample size significantly. It would be interesting to better understand the causes of this result. Figure 3 gives some good idea about the problems with the regularization scheme (parameters adjust very slowly to changes) but it is not ideal to illustrate problems with 'SW plus' and the 'Smooth model' since no NWP model upgrade happens in data set A. Wouldn't it be better to use data set B for this figure, where we can expect some adjustment during the first days/weeks of the validation period? Also, is Innsbruck the best location to illustrate the effect of a NWP model upgrade? As 'best' alpine location in this context I would consider the one that is most strongly impacted by the horizontal resolution change (this could be studied by considering changes in biases in the raw ensemble forecasts) in the ECMWF model and therefore presents a worst case scenario in terms of adjustment to a NWP model upgrade. I would encourage the authors to provide some more discussion along these lines, since NWP model upgrades have been the main argument to justify the need for reforecasts, and I am not aware of any previous study that looks at the effect of NWP model upgrades on the performance of post-processed ensemble forecasts in a quantitative way.

Minor comments:

244-245: While it's possible (even likely) that a larger slope coefficient is due to higher skill of the EPS temperature forecasts, one cannot be sure if at least to some extent the larger slope coefficient is due to an amplitude bias of the raw ensemble forecasts, i.e. the ensemble underpredicts high temperatures and overpredicts low temperatures, and increasing the slope coefficient compensates for that.
* * *

---

## Referee Comment (RC2) · Anonymous Referee #2 · 25 Oct 2019

This work investigates the effect of different types of training periods on predictive performance of postprocessing models at different types of locations (plain, alpine foreland, alpine). The presentation is concise, the aims of the work and the used methods are presented in a clear way. Especially the graphical illustration of the different types of training periods and of the situations in the considered data situations is very helpful. This comparative study is highly relevant for applications. The approaches for constructing training data presented here are all discussed in individual papers and applied to quite different situations, even based on different types of postprocessing models. Therefore, it is quite interesting to have a unified study of the effects of these training periods under the same conditions.

However, some other settings might be included in the study, and some more details in the already presented results could be interesting, see below.

General comment:

The presented study is only based on NR for the Gaussian case. It would be useful to include at least one other (NR) scenario with quite different behaviour to see whether in a case like precipitation or wind (gust) speed the results concerning the performance of the different training data sets is the same. Both precipitation and wind speed are more heavy tailed than temperature, and there can be much more localized phenomenons on maybe sub-model-grid scales. Investigation of a non-Gaussian scenario is therefore recommended.

Specific comments:

Figure 5, possible extensions: The boxplots are aggregations of all scores over the 5 stations and over all forecast horizons. It would be interesting to see these boxplots with values aggregated over the stations but for a specific forecast horizon only, e.g. exemplarily for 12h and 72h ahead. It could be interesting to see whether different forecast horizons affect the predictive performance in different ways – in conjunction with the situations (model change included or not) in datasets A, B, C.

It seems that both, SW plus and the smooth model tend to improve the forecast skill, in some scenarios in Figure 5 there is not so much difference between the two. On the contrary, the smooth model exhibits much more variation in the skill. Therefore it might be interesting to include a table or figure regarding the computation time of the different approaches. In case e.g. that the smooth model takes much more computation time than the SW and SW plus approach, then this could maybe lead to a recommendation/rule of thumb for practical use, like the more sophisticated smooth model does not provide so much more improvement than the SW plus, but has much higher computation time, so for practical use the SW plus suffices.

In that regard, the question could be addressed whether these two models indeed do not significantly differ. You might consider adding p-values of some statistical (student-t, wilcoxon, or diebold mariano) test comparing whether the average performance is significantly different or not

Technical comments:

Section 2.2.2.: You introduce the regularized sliding window approach of Scheuerer (2014). You only mention that the approach yielded better results in case of precipitation. But you do not really mention that another distribution was used in Scheuerer (2014). As your case study is only based on the normal distribution, it should be explicitly stated that the results in Scheuerer (2014) are for a non-Gaussian distribution.

Figure 3 and 4: The two validation years in data set A are both plotted in each of the panels representing a specific sliding window approach, both as dashed lines. It is really difficult to distinguish the lines belonging to the different years. Maybe you could try two different line types, and/or line thicknesses, so that one can distinguish the trajectories of the two years more easily.

Figure 5: The flat bar representing the "boxplot" for the standard sliding window approach could removed from the figure. As the standard SW approach is the reference model for the skill scores, this flat boxplot does not really provide any additional information, but it confuses at first sight

---

## Referee Comment (RC3) · Anonymous Referee #1 · 24 Nov 2019

It is interesting to see in Figs. 3 and 4 that this type of regularization seems to introduce too much inertia, i.e. the parameters only adapt with a certain delay or sometimes not at all. Have the authors tested alternative stopping criteria? A simple and obvious variant would be to perform 2 or 3 iterations on each (except the first) new day.

---

## Author Response (AR1)

To Maxime Taillardat

**Name / Email**
Moritz Lang
Moritz.Lang@uibk.ac.at

**Date**
December 10, 2019

**Revision of 'npg-2019-49'**

Dear Maxime Taillardat

We thank both reviewers for the positive and constructive feedback regarding our manuscript "**Remember the past: A comparison of time-adaptive training schemes for non-homogeneous regression**".

We have carefully revised our manuscript according to their comments and suggestions. The most substantial changes are the following:

- The main goal of the article is now more clearly stated in the manuscript. The objective is to cover a wide range of methods as proposed in the literature – rather than finding the universally best method – in order to provide guidance on strengths and weaknesses of the underlying strategies. Therefore, to show a wide spectrum of possible approaches in a unified setup, we consider typical basic applications of these training schemes and refrain from more elaborate tuning or combinations. We have adjusted the introduction (Sect. 1), the conclusion (Sect. 4) and the corresponding paragraphs in the methodology (Sect. 2.2.2) accordingly.

- We have added more information on the 2016-03-08 change in the horizontal resolution of the ECMWF EPS (cycle 41r2). This specific change was chosen to construct the data sets A-C because it is likely to affect coefficient estimates more substantially. We also now clarify that, in fact, further model changes occurred in the time periods considered but that these did not affect the horizontal resolution and hence can be expected to have much smaller effects on the coefficient estimates.

- An additional comparison of the different time-adaptive training schemes has been performed on daily precipitation sums employing a left-censored Gaussian model for post-processing. All results are very similar to the analyses for the 2 m temperature forecasts presented in the manuscript and hence nicely support the conclusions given in the paper. Therefore, we feel that it is not necessary to report these additional results in the main manuscript but we do include them in an online supplement.

On the following pages a point-to-point response to both reviewers will be given. The attached manuscript highlights the changes in the text in blue color. In addition, we have added a supplement on post-processing daily precipitation sums after the revised manuscript.

Your sincerely,

Moritz Lang
Corresponding Author

**Reviewer 1**

*This manuscript compares the effect of different schemes to compose training data for statistical post-processing methods (here: non-homogeneous regression) on the performance of the resulting forecasts. It is well written and highly relevant to operational forecasting where availability of reforecast data may be limited and the consequences of changes in the NWP model on forecast calibration must be understood in order to decide whether forecasts from an older NWP model version can be used to fit the parameters defining the post-processing model. This last point is the only one where I feel the manuscript could benefit from a more detailed discussion.*

We want to thank you for your fruitful and constructive review. We have been carefully going trough your comments to address each point individually including your general comment on NWP model changes.

Below, you can find a detailed point-to-point reply to your comments and suggestions.

*Specifically:*

*The CRPS skill scores in Fig. 5 h) suggest that the regularization scheme struggles with the adjustment to the NWP model upgrade and to the annual cycle, but also the SW plus and the smooth model have an overall neutral effect on skill even though these schemes increase the training sample size significantly. It would be interesting to better understand the causes of this result.*

Figure 5 h shows the CRPS skill scores for alpine sites for data set B. Thus, the *smooth model* is trained on the 'old EPS version' while the predictions are for the 'new EPS version' which explains its relative performance loss. This is also true for the *sliding-window plus* which is, in large parts, based on data from the 'old EPS version'. The classical *sliding-window* approach and the *regularized sliding-window* approach both adjust to the 'new EPS version' more rapidly at the same pace and, hence, show a similar predictive performance difference as for data set A (Fig. 5 g).

*Figure 3 gives some good idea about the problems with the regularization scheme (parameters adjust very slowly to changes) but it is not ideal to illustrate problems with 'SW plus' and the 'Smooth model' since no NWP model upgrade happens in data set A. Wouldn't it be better to use data set B for this figure, where we can expect some adjustment during the first days/weeks of the validation period? Also, is Innsbruck the best location to illustrate the effect of a NWP model upgrade? As 'best' alpine location in this context I would consider the one that is most strongly impacted by the horizontal resolution change (this could be studied by considering changes in biases in the raw ensemble forecasts) in the ECMWF model and therefore presents a worst case scenario in terms of adjustment to a NWP model upgrade.*

Figure 1 shows estimated coefficient paths for the weather station Altdorf for the first calendar year of the validation period of data set B. Altdorf is the station which is most strongly affected by the model change with a height difference in the model topography of 47.4 m. The first 40 days within the validation, which correspond to the period directly after the EPS change, is highlighted in pink. The variability of the coefficients within the first 40 days compared to the rest of the year are in the same order and hence no clear adjustment can be detected.

Despite the well reasoned comment, the analyses for data set B provide no further insights to the adjustment phases of the *sliding-window* and *regularized sliding-window* approaches. Hence, to restrict the presented analysis to the error sources (v) and (vi) described in Sect. 2.1 of the manuscript, we suggest to keep showing the coefficient paths for data set A in the paper. Consequently, as for data set A no EPS change has to be considered, we have kept the results for Innsbruck in the manuscript.

*I would encourage the authors to provide some more discussion along these lines, since NWP model upgrades have been the main argument to justify the need for reforecasts, and I am not aware of any previous study that looks at the effect of NWP model upgrades on the performance of post-processed ensemble forecasts in a quantitative way.*

We agree that analyzing the effects of EPS changes would be a very interesting research question on its own. We have tried to account for this issue by our study design (data set A, B, C), however, for a comprehensive perspective on this topic one would need to perform an extensive analysis on more than 15 stations. This is beyond the objective of this paper which mainly aims at presenting how time-adaptive post-processing scheme are related to each other and how these perform under specific restrictions, such as the EPS change which affects the horizontal resolution investigated in our current study. Other studies focus more on investigating the effect of model changes themselves such as, e.g., Demaeyer and Vannitsem (2019) by studying the impact of changes in a quasi-geostrophic model on post-processing.

To address your initial comment on when it is beneficial to use data from a previous NWP model for forecast calibration: As the results in our study show, the time-adaptive training schemes using multiple years of data are superior to the ones using the most recent days only, even in case of the EPS change investigated. However, this might look different

University of Innsbruck, Department of Statistics, Universitätsstrasse 15, 6020 Innsbruck, Austria
Email: Moritz.Lang@uibk.ac.at

[Figure]

**Figure 1:** *As Fig. 3 and 4 in the paper, but for station Altdorf for the first calendar year during the validation period in data set B. The first 40 days within the validation period, which correspond to the period directly after the change in the horizontal resolution of the ECMWF EPS on March 8, 2016, is highlighted in pink.*

for a different NWP model and/or future model changes and must be evaluated individually in each case. This is now explicitly stated in the conclusion of the manuscript (Sect. 4).

*It is interesting to see in Figs. 3 and 4 that this type of regularization seems to introduce too much inertia, i.e. the parameters only adapt with a certain delay or sometimes not at all. Have the authors tested alternative stopping criteria? A simple and obvious variant would be to perform 2 or 3 iterations on each (except the first) new day.*

Figure 2 shows the temporal evolution of regression coefficients exemplary for Innsbruck using three instead of only one iteration. In comparison to the *regularized sliding-window* model version presented in Fig. 3 of the manuscript, the temporal evolution of the coefficients for the modified *regularized sliding-window* approach is much more comparable to the evolution of the classical *sliding-window* approach. This increased similarity is also visible in the aggregated CRPS skill scores shown in Fig. 3. In comparison to original *regularized sliding-window* approach discussed in the manuscript, the modified *regularized sliding-window* approach has both less profound performance losses for alpine stations, and less visible performance gains for stations in the plain and foreland with the classical *sliding-window* approach as a reference.

In summary, three iterations used in the estimation process for each new day is not generally superior to a single iteration for the employed data set. To show a wide spectrum of commonly-used training schemes in a unified setup, we kept the different approaches as close as possible to the originally proposed version to show their advantages and possible disadvantages Thus, we refrain from introducing further modifications such as e.g., additional hyperparameter tuning as now stated in the conclusion (Sect. 4). In addition, we now explicitly emphasize in Sect. 2.2.3 of the paper that an increased number of iterations might be more appropriate for the *regularized sliding-window* approach depending on the employed data.

*Minor comments:*

[Figure]

**Figure 2:** *As Fig. 3 in the paper, the temporal evolution of regression coefficients is shown for the validation period in data set A for Innsbruck at forecast step $+36$ h (valid at 1200 UTC). Contrary to Fig. 3 in the paper, the modified* regularized sliding-window *approach uses three iterations in the estimation process before the optimizer is stopped. The coefficient paths are plotted for the consecutive calendar years 2014, 2015, and 2016 as dashed, dotted, and two-dashed line, respectively.*

*244-245: While it's possible (even likely) that a larger slope coefficient is due to higher skill of the EPS temperature forecasts, one cannot be sure if at least to some extent the larger slope coefficient is due to an amplitude bias of the raw ensemble forecasts, i.e. the ensemble underpredicts high temperatures and overpredicts low temperatures, and increasing the slope coefficient compensates for that.*

A very good remark! We have corrected the statement in Sect. 3.1 according to your comment.

[Figure]

**Figure 3:** *Similar to Fig. 5 in the paper, but only for the* regularized sliding-window *approaches with the classical* sliding-window *approach as a reference. The* original *version as presented in the manuscript uses a single iteration, whereas the* modified *version uses three iterations in the estimation process before the optimizer is stopped.*

**Reviewer 2**

*This work investigates the effect of different types of training periods on predictive performance of postprocessing models at different types of locations (plain, alpine foreland, alpine). The presentation is concise, the aims of the work and the used methods are presented in a clear way. Especially the graphical illustration of the different types of training periods and of the situations in the considered data situations is very helpful. This comparative study is highly relevant for applications. The approaches for constructing training data presented here are all discussed in individual papers and applied to quite different situations, even based on different types of postprocessing models. Therefore, it is quite interesting to have a unified study of the effects of these training periods under the same conditions. However, some other settings might be included in the study, and some more details in the already presented results could be interesting, see below.*

We want to thank you for your fruitful and constructive review. We have carefully addressed each of your comments and due to your suggestions we have additionally performed the comparison of the different time-adaptive training schemes for daily precipitation sums employing a non-Gaussian response distribution.

Our reply to your comments can be found on the following pages.

*General comment:*

*The presented study is only based on NR for the Gaussian case. It would be useful to include at least one other (NR) scenario with quite different behavior to see whether in a case like precipitation or wind (gust) speed the results concerning the performance of the different training data sets is the same. Both precipitation and wind speed are more heavy tailed than temperature, and there can be much more localized phenomenons on maybe sub-model-grid scales. Investigation of a non-Gaussian scenario is therefore recommended.*

[Figure]

**Figure 4:** *Similar to Fig. 3 in the paper, the temporal evolution of regression coefficients is shown for the validation period in data set A for Innsbruck at forecast step $+24$ h (valid at 0000 UTC). Contrary to Fig. 3 in the paper, regression coefficients are presented for post-processing daily precipitation sums, employing a left-censored Gaussian response distribution with a sliding window length of $80$ days.*

University of Innsbruck, Department of Statistics, Universitätsstrasse 15, 6020 Innsbruck, Austria
Email: Moritz.Lang@uibk.ac.at

[Figure]

**Figure 5:** *As Fig. 4 in this rebuttal letter, but for Hamburg at forecast step $+24\,h$ (valid at 0000 UTC).*

As the original implementation of the *regularized sliding-window* approach is based on post-processing precipitation forecasts, we have decided to present an additional analysis on post-processing daily precipitation sums. We employ the same 15 measurement sites as presented in the manuscript, and use observations and de-accumulated EPS daily precipitation sums forecasted by the ECMWF EPS. In order to remove some of the positive skewness, we follow Stauffer *et al.* (2017a) and apply a power-transformation with an ad-hoc chosen power parameter of $2$ to the observations and to every ensemble member. As an appropriate response distribution for daily precipitation sums, we employ the zero left-censored Gaussian distribution (Stauffer *et al.*, 2017a) with a sliding window length of 80 days.

Figure 4 and Fig. 5 illustrate the temporal evolution of the regression coefficients for the validation period in data set A at forecast step $+24\,h$ for Innsbruck and Hamburg, respectively. Figure 6 provides the counterpart of Fig. 5 in the manuscript, showing CRPS skill scores with the classical *sliding-window* approach as a reference. Due to the employed power-transformation, CRPS values are computed by quantile sampling with $n = 1000$; for a more detailed description compare Stauffer *et al.* (2017b).

The results for post-processing daily precipitation sums, depicted in Fig. 4–6, can be summarized as followed:

- For both Innsbruck and Hamburg, the *sliding-window* and *regularized sliding-window* approaches show very strong fluctuations in the evolution of the regression coefficient without a clear seasonal pattern comparing the consecutive years with each other (Fig. 4 and 5).

- The coefficient paths for the *sliding-window plus* approach and the *smooth model* look comparable with quite low seasonal variation in all coefficient paths. For Hamburg, the seasonal variability in the scale parameter is slightly larger than for Innsbruck (Fig. 4 and 5).

- The *sliding-window plus* and the *smooth model* approaches show the highest improvements over the classical *sliding-window* approach with a slightly better performance of the *sliding-window plus* approach for data set C in comparison to data sets A and B (Fig. 6).

All presented results are very similar to the analyses for 2 m temperature forecasts presented in the manuscript and

[Figure]

**Figure 6:** *Similar to Fig. 5 in the paper, CRPS skill scores are shown with the classical* sliding-window *approach as a reference. Contrary to Fig. 3 in the paper, regression coefficients are presented for post-processing daily precipitation sums, employing a left-censored Gaussian response distribution with a sliding window length of* 80 *days. Each box-whisker contains aggregated skill scores over the forecast steps from* +24 *h to +72 h on a 24 hourly temporal resolution and over five respective weather stations.*

support the conclusions given in the paper. Hence, we suggest to include these analyses in an online supplement in addition to the manuscript.

*Figure 5, possible extensions: The boxplots are aggregations of all scores over the 5 stations and over all forecast horizons. It would be interesting to see these boxplots with values aggregated over the stations but for a specific forecast horizon only, e.g. exemplarily for 12h and 72h ahead. It could be interesting to see whether different forecast horizons affect the predictive performance in different ways – in conjunction with the situations (model change included or not) in datasets A, B, C.*

Figure 7 shows aggregated CRPS skill scores for groups of five respective stations classified as topographically plain, mountain foreland, and alpine sites regarding solely data set A but conditional on the forecast steps from $+12$ h to $+72$ h on a $12$ hourly temporal resolution. As it can be seen, the variability of the predictive performance for the various setups is rather similar between different forecast steps. Exceptions are visible for the *smooth model* for stations located in the plains at forecast steps $+24$ h and $+48$ h (0000 UTC) and for stations located in the foreland at forecast steps $+36$ h and $+60$ h (1200 UTC). While the variance for the plain sites increases and the predictive performance slightly decreases, the performance for the foreland sites show the exact opposite.

As the variability between the different forecast steps is overall within a reasonable range and does not show any distinct pattern, we think that Fig. 7 provides no significantly new insights to the research question of the manuscript.

*It seems that both, SW plus and the smooth model tend to improve the forecast skill, in some scenarios in Figure 5 there is not so much difference between the two. On the contrary, the smooth model exhibits much more variation in the skill. Therefore it might be interesting to include a table or figure regarding the computation time of the different approaches. In case e.g. that the smooth model takes much more computation time than the SW and SW plus approach, then this could maybe lead to a recommendation/rule of thumb for practical use, like the more sophisticated smooth*

[Figure]

**Figure 7:** *As Fig. 5 in the paper, CRPS skill scores are shown with the classical* sliding-window *approach as a reference. Contrary to Fig. 5 in the paper, all scores are shown only for data set A but conditional on the forecast steps from $+12\,h$ to $+72\,h$ on a $12$ hourly temporal resolution.*

*model does not provide so much more improvement than the SW plus, but has much higher computation time, so for practical use the SW plus suffices.*

The computation time for the various sliding-window approaches is in the order of seconds, whereas the estimation of the *smooth model* takes a few minutes. The latter is however estimated using MCMC sampling, which allows drawing inferential conclusions about the selected terms but is not mandatory for practical use.

In addition, for the sliding-window approaches the NR models must be re-estimated every day, whereas the same *smooth model* is valid for several years. We have included these times in the end of Sect. 2.3.2, but want to point out that these are only valid for the employed estimation software listed in the section "code availability".

*In that regard, the question could be addressed whether these two models indeed do not significantly differ. You might consider adding p-values of some statistical (student-t, wilcoxon, or diebold mariano) test comparing whether the average performance is significantly different or not.*

The purpose of the evaluation is to reveal patterns in the performance of the different strategies and to build a better awareness of possible constraints of the various methods, rather than to evaluate if the performance differences are significant. As summarized in Sect. 4 of the manuscript, all four training schemes have their advantages in particular applications; at the end it's up to the user to select the appropriate training-scheme for his/her specific application. Thus, we have have decided not to include an additional evaluation of the predictive performances of the different methods as we think that it may distract the readers from the main objective of the article.

*Technical comments:*

*Section 2.2.2.: You introduce the regularized sliding window approach of Scheuerer (2014). You only mention that the approach yielded better results in case of precipitation. But you do not really mention that another distribution was used in Scheuerer (2014). As your case study is only based on the normal distribution, it should be explicitly stated that the results in Scheuerer (2014) are for a non-Gaussian distribution.*

Thank you for pointing that out. We agree and have rephrased "post-processing precipitation amounts employing a left-censored generalized extreme value distribution" in Sect. 2.2.2.

*Figure 3 and 4: The two validation years in data set A are both plotted in each of the panels representing a specific sliding window approach, both as dashed lines. It is really difficult to distinguish the lines belonging to the different years. Maybe you could try two different line types, and/or line thicknesses, so that one can distinguish the trajectories*

University of Innsbruck, Department of Statistics, Universitätsstrasse 15, 6020 Innsbruck, Austria
Email: Moritz.Lang@uibk.ac.at

*of the two years more easily.*

Thank you for pointing out this possible confusion. For clarity, we now use different line types for the consecutive calendar years in both Fig. 3 and 4.

*Figure 5: The flat bar representing the "boxplot" for the standard sliding window approach could removed from the figure. As the standard SW approach is the reference model for the skill scores, this flat boxplot does not really provide any additional information, but it confuses at first sight*

We agree that the flat box-whiskers provide no additional information. We had included these as a visible reference, but this has apparently not added to the clarity of this figure. Hence, we have removed the reference from Fig. 5 in the manuscript.

**\*References**

Demaeyer J, Vannitsem S (2019). "Correcting for Model Changes in Statistical Post-Processing – An approach based on Response Theory." *Nonlinear Processes in Geophysics Discussions*, **2019**, 1–27. doi:10.5194/npg-2019-57.

Stauffer R, Mayr GJ, Messner JW, Umlauf N, Zeileis A (2017a). "Spatio-Temporal Precipitation Climatology over Complex Terrain Using a Censored Additive Regression Model." *International Journal of Climatology*, **37**(7), 3264–3275. doi:10.1002/joc.4913.

Stauffer R, Umlauf N, Messner JW, Mayr GJ, Zeileis A (2017b). "Ensemble Postprocessing of Daily Precipitation Sums over Complex Terrain Using Censored High-Resolution Standardized Anomalies." *Monthly Weather Review*, **145**(3), 955–969. doi:10.1175/MWR-D-16-0260.1.

[revised manuscript text omitted]

**Supplement A: Post-processing of daily precipitation sums**

**Moritz N. Lang**
Universität Innsbruck

**Sebastian Lerch**
Karlsruher Institut für Technologie

**Georg J. Mayr**
Universität Innsbruck

**Thorsten Simon**
Universität Innsbruck

**Reto Stauffer**
Universität Innsbruck

**Achim Zeileis**
Universität Innsbruck

**Abstract**

The study presented in the manuscript "Remember the past: A comparison of time-adaptive training schemes for non-homogeneous regression" compares three widely-used training approaches with the classical sliding-window model for the application of post-processing near-surface air temperature forecasts across Central Europe. While the normal distribution is typically employed for post-processing air temperatures, this supplement extends the study by post-processing daily precipitation sums using a zero left-censored Gaussian distribution. Despite the different characteristics of daily precipitation sums and the alternative response distribution, the results are very similar to the ones for $2\,\mathrm{m}$ temperature forecasts and, hence, nicely support the conclusions given in the paper.

*Keywords*: Non-homogeneous regression, training data, sliding training window, post-processing, regression splines, ensemble forecasts, daily precipitation sums.

**1. Introduction**

This supplement extends the comparison study presented in the main manuscript by performing the same evaluation for daily precipitation sums employing an alternative response distribution.

As in the main manuscript the temporal evolution of the estimated coefficients is shown for two stations with different site characteristics followed by the analysis of the predictive performance. The results of all training schemes is evaluated in terms of the continuous ranked probability score (CRPS) conditional on the three data sets with and without the change in the horizontal resolution of the ensemble prediction system (EPS) on March, 8, 2016, as well as grouped for stations classified as topographically plain, mountain foreland, and alpine site. As a reminder, the different training-schemes are briefly summarized as follows:

- *Sliding-window:* The classical *sliding-window* approach as introduced by Gneiting, Raftery, Westveld III, and Goldman (2005) uses solely the $n$ most recent days prior to the day of interest as training data to estimate the statistical models.

- *Regularized sliding-window:* The regularized adaption of the classical *sliding-window*

approach stabilizes the estimation based on early stopping in statistical learning. In this study, it is applied in its original version where the coefficients of the previous day are used as starting values and the optimizer is stopped after a single iteration (Scheuerer 2014).

- *Sliding-window plus:* In order to stabilize the coefficient estimates and to address seasonal effects, data from previous years are additionally included in the training data set. In contrast to the classical *sliding-window* approach, the most recent $n$ days prior to estimation and a respective $(2n + 1)$ days interval centered around the day of interest over the previous years available are used to estimate the coefficients (Vogel, Knippertz, Fink, Schlueter, and Gneiting 2018; Möller, Spazzini, Kraus, Nagler, and Czado 2018).

- *Smooth model:* Rather than adapting the training data set, the *smooth model* makes use of all historical data in combination with cyclic regression splines which allows the coefficients to smoothly evolve over the year.

In comparison to the main manuscript, to account for potentially long periods without precipitation at a specific site, we use a training length of $n = 80$ days in the sliding-window approaches for post-processing daily precipitation sums. This is in the order of common choices in the literature (e.g., Baran and Nemoda 2016) and exactly two times the length employed for post-processing 2 m temperature forecasts as presented in the main manuscript.

**2. Methodology and data**

To account for non-negative values, the large fraction of zero observations, and the heavy-tailed distribution of precipitation, we proceed as proposed by Stauffer, Mayr, Messner, Umlauf, and Zeileis (2017a): We power-transform observed and modeled daily precipitation sums (member-by-member) with an ad-hoc chosen power parameter of 2. This transformed precipitation $y$ can be assumed to follow a zero left-censored Gaussian distribution $\mathcal{N}_0$,

$$y \sim \mathcal{N}_0(\mu, \sigma). \tag{1}$$

As in the manuscript, the two distribution parameters location $\mu$ and scale $\sigma$ are expressed by the ensemble mean $m$ and ensemble variance or standard deviation $s$, respectively:

$$\mu = \eta_\mu = \beta_0 + \beta_1 \cdot m, \tag{2}$$
$$\log(\sigma) = \eta_\sigma = \gamma_0 + \gamma_1 \cdot s, \tag{3}$$

with $\beta_\bullet$ and $\gamma_\bullet$ being the corresponding intercept and slope coefficients.

As covariates, we employ the ensemble mean $m$ and the ensemble standard deviation $s$ of bilinearly interpolated power-transformed daily precipitation sum forecasts issued by the global 50-member EPS of the European Centre for Medium-Range Weather Forecasts (ECMWF). Ensemble forecasts and corresponding observations are considered at 15 measurement sites located across Austria, Germany, and Switzerland. The data comprises three groups of five stations located either in plains, mountain foreland, and within mountainous terrain. An overview of the study area is provided in the main manuscript in Fig. 1. For training and validation, we assess forecast steps from +24 h to +72 h ahead on a 24 hourly temporal resolution for the EPS run initialized at 0000 UTC and use the same data period employed in the manuscript, from March 8, 2010 to March 7, 2019.

[Figure]

Figure 1: Temporal evolution of regression coefficients for the validation period in data set A, cf. Fig. 2 in the main manuscript, for Innsbruck at forecast step +24 h (valid at 0000 UTC). The coefficient paths are shown for the coefficients $\beta_0$ (a–c) and $\beta_1$ (d–f) in the location parameter $\mu$, and for the coefficients $\gamma_0$ (g–i) and $\gamma_1$ (j–l) in the scale parameter $\sigma$ based on the *sliding-window*, *regularized sliding-window*, and *sliding-window plus* approach (dashed, from left to right) compared to the *smooth model* approach (solid line). The coefficient paths are plotted for the consecutive calendar years 2014, 2015, and 2016 as dashed, dotted, and two-dashed line, respectively. The grey shading represents the 95% credible intervals of the coefficients in the *smooth model* based on MCMC sampling.

**3. Results**

In comparison to the main manuscript, the presented results compare the performance of the different time-adaptive training schemes for post-processing daily precipitation sums. Figure 1 and Fig. 2 illustrate the temporal evolution of the estimated coefficients shown for two stations representative for one measurement site in the plains (Hamburg) and one in mountainous terrains (Innsbruck). Figure 3 shows the predictive performance of the training schemes evaluated for three groups of stations with different site characteristics and in terms of the CRPS conditional on the three data sets with and without the change in the horizontal resolution of the EPS, as defined in Sect. 2.3.2 of the main manuscript. Due to the employed power-transformation, CRPS values are computed by quantile sampling with $n = 1000$; for a more detailed description compare Stauffer, Umlauf, Messner, Mayr, and Zeileis (2017b).

[Figure]

Figure 2: As Fig. 1, but for Hamburg at forecast step +24 h (valid at 0000 UTC).

The results for post-processing daily precipitation sums, depicted in Fig. 1–3, can be summarized as followed:

- For both Innsbruck and Hamburg, the *sliding-window* and *regularized sliding-window* approaches show very strong fluctuations in the evolution of the regression coefficient without a clear seasonal pattern comparing the consecutive years with each other (Fig. 1 and 2).

- The coefficient paths for the *sliding-window plus* approach and the *smooth model* look comparable with quite low seasonal variation in all coefficient paths. For Hamburg, the seasonal variability in the scale parameter is slightly larger than for Innsbruck (Fig. 1 and 2).

- The *sliding-window plus* and the *smooth model* approaches show the highest improvements over the classical *sliding-window* approach with a slightly better performance of the *sliding-window plus* approach for data set C in comparison to data sets A and B (Fig. 3).

**4. Conclusion**

This supplement provides a comparison evaluating the different time-adaptive training schemes for post-processing daily precipitation sums. To account for non-negative values, the large

[Figure]

Figure 3: CRPS skill scores clustered into groups of stations located in the plain, in the mountain foreland near the Alps, and within mountainous terrain and for the out-of-sample validation periods according to the different data sets: Data set A without the change in the horizontal resolution of the EPS, data set B with the EPS change in between the training and the validation data sets, and data set C with the EPS change withing training data (cf. Fig. 2 in the main manuscript). Compared are the different time-adaptive training schemes specified in Sect. 2 with the classical *sliding-window* approach as a reference; note that 'sliding-window' is abbreviated as SW in the figure. Each box-whisker contains aggregated skill scores over the forecast steps from +24 h to +72 h on a 24 hourly temporal resolution and over five respective weather stations (cf. Fig. 1 in the main manuscript). Skill scores are in percent, positive values indicate improvements over the reference.

fraction of zero observations, and the strongly positively skewed characteristics of daily precipitation sums, we employ a power-transformation to both the observations and to each ensemble member, and use the zero left-censored Gaussian distribution in the framework of non-homogeneous regression (Stauffer *et al.* 2017a).

Despite the different characteristics of daily precipitation sums and the alternative response distribution, the results are very similar to the ones for 2 m temperature forecasts presented in the main manuscript. This shows that the findings for 2 m temperature can also be transferred to other quantities using different model assumptions and, hence, nicely support the conclusions given in the paper.

**Affiliation:**

Moritz N. Lang
Department of Statistics
Department of Atmospheric and Cryospheric Sciences
Universität Innsbruck
6020 Innsbruck, Austria
E-mail: `moritz.lang@uibk.ac.at`